# Using mobility data in the design of optimal lockdown strategies for the COVID-19 pandemic

**Ritabrata Dutta**[1]*, **Susana N. Gomes**[2], **Dante Kalise**[3¤], **Lorenzo Pacchiardi**[4]

**1** Department of Statistics, Warwick University, Coventry, United Kingdom, **2** Department of Mathematics, Warwick University, Coventry, United Kingdom, **3** School of Mathematical Sciences, University of Nottingham, Nottingham, United Kingdom, **4** Department of Statistics, University of Oxford, Oxford, United Kingdom

¤ Current address: Department of Mathematics, Imperial College London, London, United Kingdom
* ritabrata.dutta@warwick.ac.uk

**Data Availability Statement:** All the data used in this paper are publicly available from NHS England (https://www.england.nhs.uk/statistics/statistical-work-areas/covid-19-daily-deaths/), Public Health England (https://www.gov.uk/government/

## Abstract

A mathematical model for the COVID-19 pandemic spread, which integrates age-structured Susceptible-Exposed-Infected-Recovered-Deceased dynamics with real mobile phone data accounting for the population mobility, is presented. The dynamical model adjustment is performed via Approximate Bayesian Computation. Optimal lockdown and exit strategies are determined based on nonlinear model predictive control, constrained to public-health and socio-economic factors. Through an extensive computational validation of the methodology, it is shown that it is possible to compute robust exit strategies with realistic reduced mobility values to inform public policy making, and we exemplify the applicability of the methodology using datasets from England and France.

## Author summary

In many countries, the COVID-19 pandemic has revealed a gap between public policy making and the use of advanced technological tools to inform such a process. In the big data era, decisions concerning the implementation of quarantines and travel restrictions are still being taken based on incomplete public health data, despite the myriad of information our society provides in real time, such as mobility data, commuting network structures, and financial patterns, to name a few. To advance towards an effective data-driven, quantitative policy making, we propose a computational framework where a predictive epidemiological model is fitted by feeding both public health and Google mobility data. The resulting model is then used as a basis for designing mobility reduction strategies which are optimised taking into account both the healthcare system capacity, and the economic impact of an extended lockdown. For the COVID-19 pandemic in England and France, we show that it is possible to design lockdown policies allowing a partial return to workplaces and schools, while maintaining the epidemic under control.

publications/slides-and-datasets-to-accompany-coronavirus-press-conference-25-may-2020), Santé publique France (https://www.santepubliquefrance.fr/dossiers/coronavirus-covid-19) and Google (https://www.google.com/covid19/mobility/). Further, all the data and codes used in our paper can be found in our public github repository: https://github.com/OptimalLockdown/.

**Funding:** RD, DK and SG were supported by EPSRC grant "COVID19: Optimal Lockdown", reference EP/V025899/1. DK was supported by a public grant as part of the Investissement d'avenir project, reference ANR-11-LABX-0056-LMH; LabEx LMH. DK was further supported by the EPSRC grants EP/V04771X/1, EP/T024429/1. SG is supported by the Leverhulme Trust through the Early Career Fellowship ECF- 2018-536. LP is supported by the EPSRC and MRC through the OxWaSP CDT programme (EP/L016710/1). The funders had no role in study design, data collection and analysis, decision to publish, or preparation of the manuscript.

**Competing interests:** The authors have declared that no competing interests exist.

# Introduction

The COVID-19 pandemic has put quantitative decision-making methods (and the lack thereof) in the spotlight. Designing informed non-pharmaceutical intervention strategies (NPIS) to mitigate the pandemic effects has been a controversial issue worldwide. In particular, the planning of effective lockdown policies and their posterior lifting based on real-time data still remains a largely open problem. A vast amount of research efforts has been dedicated to model the COVID-19 pandemic focusing on the various aspects of the system dynamics, such as estimating the value of the basic reproductive number [1, 2], evaluating the effect of containment measures and travel restrictions [2–6], assessing the effect of age on the transmission and severity of the disease [1, 7] and estimating the impact on Health Services [8]. It is remarkably hard, if not impossible, to capture every aspect of this complex phenomenon in an integrated and computationally tractable mathematical model. With this in mind, our goal in the present work is to study the dynamics of COVID-19 spread by integrating a dynamical epidemiological model with mobile phone data from the population. Such an adjusted model yields an accurate account of the real displacement of the population between different locations (e.g. workplaces, schools, etc.) during the pandemic, and serves as the basis for determining lockdown and exit strategies which are optimised according to public health and socio-economic constraints. We would like to stress here that, in order to define the model dynamics, we have considered some assumptions (on the contact matrices, the connection between mobility data and contacts, the initialisation of the compartments and the definition of the cost functional) which may not be easily tested in real a life scenario.

## Related literature

Previous works have modelled the impact of NPIS using extensions of the classical Susceptible-Infected-Removed (SIR) model [9] with the inclusion of compartments corresponding to asymptomatic population which are either exposed to infections but not yet infectious, or infected and infectious [10]. Given the current knowledge of the COVID-19 disease, this constitutes a complete description of the possible states.

Among the works based on the aforementioned state space representation, [5], one of the main inspirations behind our work, uses an age-structured model to quantify the effect of control measures imposed in Wuhan, China and concludes that there exists a large potential on the use of NPIS for mitigating the COVID-19 pandemic. Furthermore, the authors recommend a gradual relaxation strategy in comparison to an early lifting of the imposed lockdown measures to avoid possible second and third waves of the pandemic. Using a stochastic modification of a compartmental model without age structure, [2] reaches similar conclusions and quantifies the effectiveness of lockdown measures by estimating the reproduction number, which decreased from a median value of 2.35 before travel restrictions were imposed, to 1.05 one week after the implementation of travel restrictions.

In the context of resorting to optimal control methods to determine a lockdown policy, [11] compares a switching on-off strategy with a two-stage release from quarantine (with part of the population released first, and the others later). The authors consider a threshold-based sanitary cost functional aiming at releasing the largest possible population without exceeding the availability of hospital beds, and their conclusion is to favour the second strategy. [12] consider multiple control levers, such as the number of tests (both virologic and anti-body tests) and the increase of ICU beds in addition to the reduction of social contacts, and uses a cost functional involving both economic and sanitary costs. The authors suggest an optimal lockdown policy which involves a quick and strong isolation, followed by a large increase in the number of tests.

## Our contribution

While the qualitative modelling and control concepts of the aforementioned works are aligned with the epidemiological literature, we remark that most of these papers use parameter values collected from previous works or estimated using either different models or from clinical knowledge. Hence, they cannot be directly applied to populations with different spatio-temporal patterns. In contrast, the main goal of this paper is to propose a framework which calibrates the model using epidemiological and real-time mobility data from a specific population, measured by Google Mobility through Android devices, and computes an optimal lockdown strategy for that population at any point of the pandemic. To achieve this, we combine parameter estimation for an epidemiological model with a subsequent optimal control step. Secondly, our framework includes the computation of the optimal lockdown policy in a nonlinear model predictive control framework, leading to a robust feedback protocol which allows not only real-time adaptation of the release strategy but also a partial lockdown, as opposed to a switching on/off strategy which can be too restrictive.

## Methodological summary

Our epidemiological model simulates the transmission dynamics of COVID-19 spread in the English and French populations using anonymised data on the reduction of the population mobility collected through smartphones and released by Google. Further, exploiting Approximate Bayesian Computation (ABC) [13], we calibrate this model using data on daily deaths and the number of people in hospitals with COVID-19 released from public health authorities (Public Health England (PHE) and the National Health Service (NHS) in England, and Santé publique France (SpF) in France) [14–17]. In ABC, we assume a prior distribution for each parameter value and, given a dataset with an inherent observation noise, we obtain a non-parametric estimate of the joint probability distribution of the parameter values. This allows us to estimate parameters of our model for the specific cases of England and France, as opposed to inheriting parameters from epidemiological models from other countries. Some key attributes of our model are the inclusion of age-dependent transition probabilities between the different compartments which are also estimated from data, as well as age-dependent social distancing, and the use of Google mobility data to quantify the effect of social distancing measures in reality. Having calibrated our model, we design a lockdown strategy which is differentiated according to social contact categories including schools, work, and others. This allows us to assign a different economic penalty for each one of them. Borrowing a leaf from optimal control theory, we synthesize an optimised lockdown and exit strategy which minimizes the number of COVID-19-related casualties in the population, but also takes into account economic constraints. In order to perform this task, we quantify the relation between the decrease in social contacts with the reduction in the population mobility and optimise with respect to the latter, which is an effectively measurable quantity compared to an abstract decrease in social contacts. A methodological summary is depicted in Fig 1, illustrating the interaction among the different building blocks of our approach. It is important to note that the proposed methodology transcends the design of lockdown strategies for the COVID-19 pandemic, and can be applied for more general epidemiological models, different datasets, and a variety of control objectives. What is fundamental in our approach is the existence of a dynamical model, the assimilation of data for the optimal estimation of model parameters and uncertainties, and the optimization of an external input action to control the system towards a desired state.

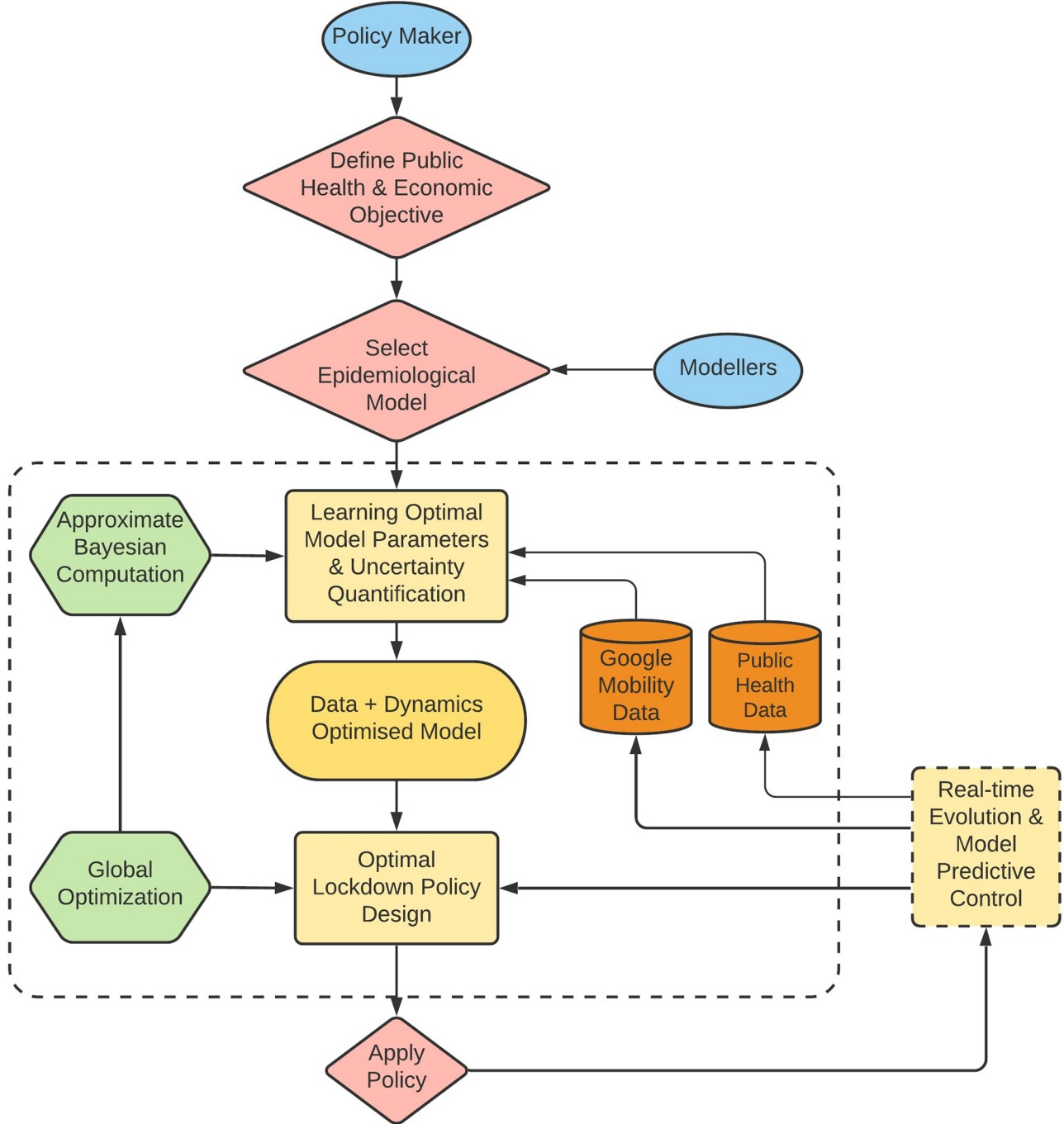

**Fig 1. Flow diagram of the data-driven approach for the synthesis of optimal lockdown policies.** The initial step consists of a policy maker defining a performance measure based on sanitary and economic objectives, and a modeller selecting a consistent generic epidemiological model. Then, public healthcare/mobility data is used in conjunction with Approximate Bayesian Computation to calibrate the dynamical model and determine the degree of uncertainty in the model parameters. This assists the formulation of an optimal control problem where the original sanitary + economic performance measure is optimized constrained to the calibrated epidemiological model. An optimal lockdown policy is then computed via global optimization techniques, and the final output is an optimal lockdown policy. The optimised lockdown is then applied and its real-time effects can be sensed through public data, which fed back into the learning and optimization framework for re-computation and update.

## Epidemiological model using Google mobility

Our epidemiological model is developed in order to exploit data on change of mobility (in our case, provided by Google) and information on the social contacts patterns at different types of locations, such as schools or workplaces (eg. estimated by the BBC Pandemic project [18] for the UK and the POLYMOD study [19] for 152 countries in the world). In this manuscript, we will focus on explaining the model for England and the methodology will be illustrated for both England and France; however, we stress that the methodology can be extended to consider any other country, and, in fact, to other epidemiological models, where suitable data is available. In this Section, we first explain the fundamental dynamic properties considered in this model and then explain how these two data sources are used.

### Model dynamics

Assuming a well-mixed population (namely, each person has the same probability of interacting with any other person in the population), we consider a compartmental model [20], splitting the population in different *compartments* representing different states of the infection. Although this is extremely simplifying, compartmental models under this assumption are widely used to describe the dynamics of epidemics over large populations. Our model considers the following compartments:

- Susceptible ($S$), meaning people who did not have any contact with the infection,

- Exposed ($E$) to the infection, but not yet infectious,

- Infected SubClinical ($I^{SC}$, split in $I^{SC1}$ and $I^{SC2}$), not needing medical attention,

- Infected Clinical ($I^C$, split in $I^{C1}$ and $I^{C2}$), needing medical attention,

- Recovered ($R$), which we assume are resistant to a new infection,

- Deceased ($D$).

As strong evidence towards the age-dependent severity of COVID-19 has been observed in previous research works [1, 7], we consider age-stratification of all of the states along 5 age groups: 0–19, 20–39, 40–59, 60–79, 80+, hence we will use the notation $E_i$ to denote the Exposed population in the $i$-th age group, and similarly for the other states. The model will assume that all the age groups are susceptible to the infection in the same way, but that the severity is strongly dependent on the age of the patient through age-dependent probabilities of necessity of hospitalization ($\rho_i$) and death if hospitalized ($\rho_i'$) for the $i$-the age group.

Another key assumption of our model is that when a patient is hospitalized and diagnosed, they are isolated and therefore not able to spread the infection. To reflect this scenario, we assume that from the exposed state and after some incubation period, all patients will become sub-clinical $I^{SC}$, in which state they are infectious. Afterwards, some of them will recover ($R$) and others will need clinical help ($I^C$); we model this by splitting $I^{SC}$ into two categories: the ones recovering straightaway ($I^{SC2}$) and the ones in need of clinical care ($I^{SC1}$). The split happens with an age-dependent probability $\rho_i$. After some time, people in $I^{SC1}$ will go to hospital, therefore moving to the $I^C$ state; similarly as before, the latter state is split in two categories according to the final outcome: the ones in $I^{C1}$ will decease ($D$) after some time, while the ones in $I^{SC2}$ will recover ($R$). This split is again described by an age-dependent probability, which we denote as $\rho_i'$. A visualization of the dynamics is given in Fig 2.

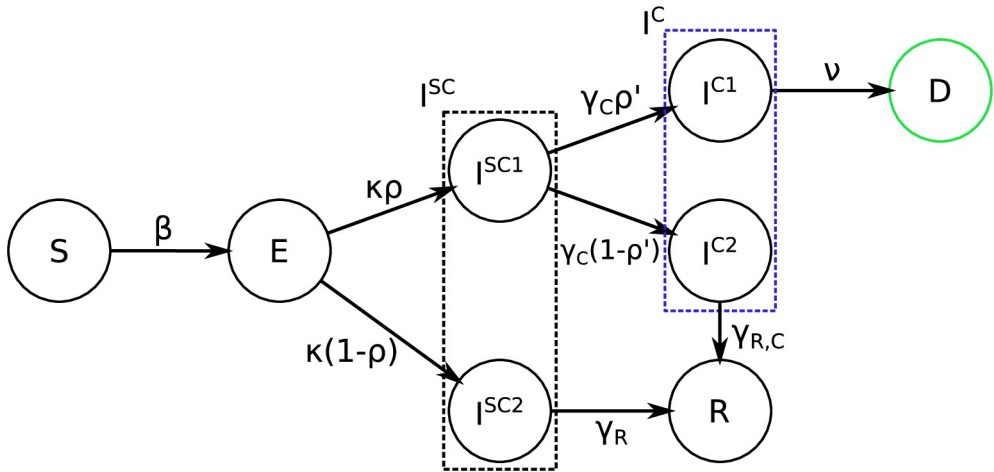

**Fig 2. Graphical representation of the model, for each age group.** The green color represents a compartment that is observed independently for each age group, while blue represents a compartment whose sum across age groups is observed.

Mathematically, this can be described using the following system of ordinary differential equations (ODEs):

$$\frac{dS_i}{dt} = -\beta S_i \sum_j C_{ij} \frac{I_j^{SC}}{N_j} \tag{1}$$

$$\frac{dE_i}{dt} = \beta S_i \sum_j C_{ij} \frac{I_j^{SC}}{N_j} - \kappa E_i \tag{2}$$

$$\frac{dI_i^{SC_1}}{dt} = \rho_i \kappa E_i - \gamma_C I_i^{SC_1} \tag{3}$$

$$\frac{dI_i^{SC_2}}{dt} = (1 - \rho_i)\kappa E_i - \gamma_R I_i^{SC_2} \tag{4}$$

$$\frac{dI_i^{C_1}}{dt} = \rho_i' \gamma_C I_i^{SC_1} - v I_i^{C_1} \tag{5}$$

$$\frac{dI_i^{C_2}}{dt} = (1 - \rho_i')\gamma_C I_i^{SC_1} - \gamma_{R,C} I_i^{C_2} \tag{6}$$

$$\frac{dR_i}{dt} = \gamma_{R,C} I_i^{C_2} + \gamma_R I_i^{SC_2} \tag{7}$$

$$\frac{dD_i}{dt} = v I_i^{C_1}, \tag{8}$$

where $I_j^{SC} = I_j^{SC1} + I_j^{SC2}$ and $C$ is the contact matrix representing the frequency of contacts between different age groups [18], where each element $C_{ij}$ represents the average daily number

of contacts a person in age group $i$ has with people in age group $j$. To simulate from the model, the ODEs (Eqs (1)–(8)) are integrated using a 4th-order Runge Kutta integrator, with a time-step $dt = 0.1$ days; the dynamics is started on the 1st of March.

**Influence of mobility on the dynamics.** The contact matrices are a crucial component defining our model dynamics, and were estimated in the POLYMOD study [19] for a large set of countries (among which England and France). Specifically for the contact matrices for England, the findings of the more recent BBC Pandemic project [18] were integrated using the procedure described in [18]. Note that [18] provided contact matrices for the whole UK, We assume that the contact matrix for England to be the same as that for the whole of the UK. Finally, the age groups considered in these studies (namely, 5 year bands) are finer than the ones we consider in the present work; we therefore aggregate the data to make contact matrices suit our needs. The $(i, j)$–th entry of this contact matrices at different locations (eg. home, workplace, school and other locations) represent the amount of daily contacts an individual in age group $i$ has with individuals from age group $j$ in different settings (see Fig 3). Before the lockdown, the total contact matrix is simply the sum of the contributions of these different locations:

$$C = C^{home} + C^{work} + C^{school} + C^{other}.$$

However, the introduction of lockdown measures lead to considerable change to people's social activity and mobility; we model this by introducing a set of multipliers (for each age group and for each of the locations) which will represent the change in the number of social contacts:

$$C_{i,j} = \alpha_i^{home} C_{i,j}^{home} + \alpha_i^{work} C_{i,j}^{work} + \alpha_i^{school} C_{i,j}^{school} + \alpha_i^{other} C_{i,j}^{other}, \tag{9}$$

where $\alpha_i^{school}, \alpha_i^{other}, \alpha_i^{work}$, represents the change of social contacts for age group $i$ in the locations *school, other, and work*. These multipliers are a function of time and not easily accessible. Instead, it is rather easy to measure the reduction of people's mobility towards the different locations; we choose then to express the $\alpha$'s as a function of the mobility values provided by Google, as explained in the next paragraph.

**Mobility data** is collected by Google to reflect the reduction of the population mobility during lockdown for each country, by following the movements of Android phones; anonymised

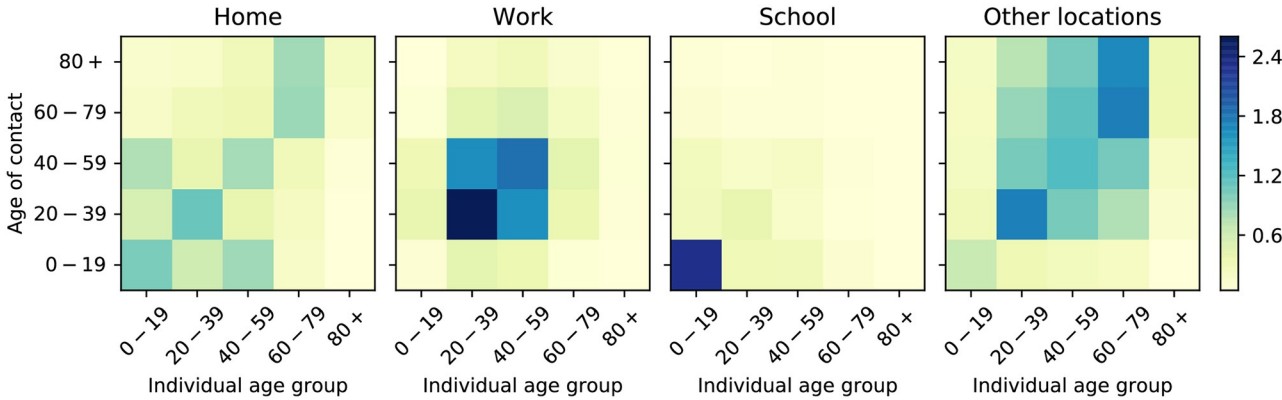

**Fig 3. Contact matrices at different locations in the UK for the age groups used in the present study (0–19, 20–39, 40–59, 60–79, 80+); these are obtained by aggregating and combining the contact matrices for 5-year bands provided by [18] and [19].**

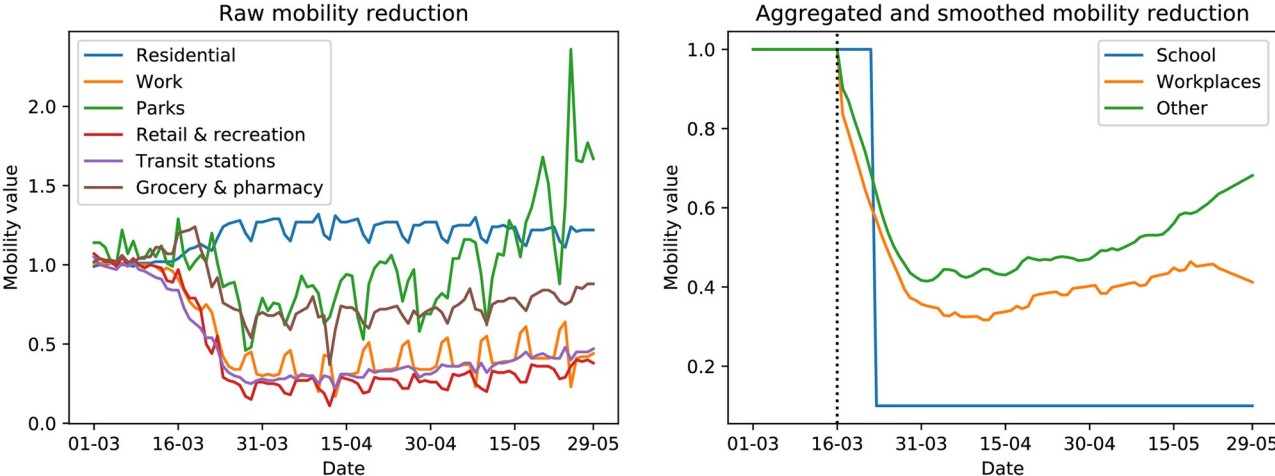

**Fig 4. Raw and elaborated mobility data in the UK.** In the raw mobility data, which is scaled with respect to a baseline value representing average mobility in the months prior to the pandemics, a strong weekly seasonality is present, which we mostly removed using a Savitzky–Golay filter. Moreover, we note that the mobility towards "residential" (which is not used in our analysis) locations is larger during the lockdown months than before, as people spend more time in their homes. Finally, we note the very large increase in people's mobility towards parks around the end of the winter season. That contribution is however only one of the components in our aggregated $m^{other}$ mobility value, so the latter does not increase that abruptly.

data is publicly available [21]. Similar datasets can be retrieved directly from other sources, such as mobile phone companies [22]. This dataset captures mobility towards the following locations: "residential", "workplaces", "parks", "retail and recreations", "transit stations" and "grocery and pharmacy", which we denote respectively as $m^{residential}$, $m^{work}$, $m^{parks}$, $m^{retail}$, $m^{transit}$, $m^{grocery}$; the changes in mobility are reported with respect to the baseline values prior to introduction of lockdown measures. As can be seen in the left panel of Fig 4, a strong weekly periodicity is present in this data; we therefore use a Savitzky–Golay filter [23] to remove it.

We moreover combined the mobility values $m^{parks}$, $m^{retail}$, $m^{transit}$, $m^{grocery}$ in order to obtain an aggregated value for the reduction of mobility towards "other locations": $m^{other} = 0.1 \cdot m^{parks} + 0.3 \cdot m^{retail} + 0.3 \cdot m^{transit} + 0.3 \cdot m^{grocery}$. Even though the numerical values of the weights are arbitrary, this choice is motivated by our observation that the value of the different contributions of the mobility data is quite similar; we also attribute a smaller value to "parks" as people get less in strict contact with each other there with respect to "retail", "transit" or "grocery" locations.

As no data with regards to schools were provided, we fixed the value of $m^{school}$ to be 0.1 from the day schools and universities were closed except for children of essential workers (23rd of March [24] for England, 16th March for France [25]). The mobility data obtained after the aggregation and smoothing operations described above is presented in the right panel of Fig 4.

In order to connect the reduction in mobility to the reduction in the number of contacts, we proceed in the following way: first we assume that the number of residential contacts stays constant; in fact, we expect the behavior of people at home not to differ too much with respect to what it was prior to the introduction of lockdown measures; for our model, this amounts to fixing $\alpha_i^{home}(t) = 1$, $\forall t$, $\forall i$. With regards to the remaining contributions to the total number of contacts, we expect data on mobility reduction to be representative of the subset of the population which mostly uses smartphones, which is likely to be younger than 60 years old. Moreover, a large part of the population older than 60 years old does not take part in work or school

activities. Motivated by these arguments, we define the values of $\alpha_i^{school}$, $\alpha_i^{other}$ and $\alpha_i^{work}$ separately for population below and above 60. Specifically, we assume the following for age groups below 60 years old (age group index 1,2,3):

$$\begin{cases} \alpha_i^{school}(t) & = \alpha_{123} \cdot m^{school}(t), \\ \alpha_i^{other}(t) & = \alpha_{123} \cdot m^{other}(t), \qquad \text{for } i \in \{1,2,3\} \\ \alpha_i^{work}(t) & = \alpha_{123} \cdot m^{work}(t), \end{cases} \qquad (10)$$

where we made the dependence on time explicit in order to highlight that $\alpha_{123} \in [0,1]$ is a time-independent scalar, which is an additional parameter in our model; this amounts to assuming that the reduction in the number of contacts is due to a combination of reduced mobility and increased awareness of people, for instance by maintaining social distancing. For people above 60 years old (age group index 4 and 5) we assume instead that the reduction in contacts stays constant since the introduction of lockdown measures and that such reduction is equally distributed across the different categories (as the contacts for *work* and *school* will be relatively few):

$$\alpha_4^{school}(t) = \alpha_4^{other}(t) = \alpha_4^{work}(t) = \alpha_4, \quad \alpha_5^{school}(t) = \alpha_5^{other}(t) = \alpha_5^{work}(t) = \alpha_5, \qquad (11)$$

where $\alpha_4, \alpha_5 \in [0,1]$ are time-independent scalars, which are parameters in our model as well. This latter assumption is motivated by the fact that such part of the population is more susceptible to the disease, so that the official advice will be for them to be as isolated as possible throughout the epidemics.

We initialise our implementation of the model dynamics on the 1st of March and we fix the contact matrix to be the standard one relative to the country until the introduction of lockdown measures (which we assume to be on the 18th March, i.e. two days after the UK government advised people to self-isolate [26] and one day after the French government banned all except essential journeys [27]); from that day onward, we use the contact matrix obtained from Eq (9), by fixing $\alpha^{home} = 1$ and obtaining the values for the other $\alpha$'s by Eqs (10) and (11).

**Intitialization and model parameters.** At the beginning of the dynamics, most of the population is in the $S$ state, except for a small number of individuals which seed the infection. We therefore assume that some people were already infected on the 1st of March and we denote that number as $N^{in}$; this number is split across the different categories and age groups in the following way:

- First, the total number of infected population is spread across the age groups with the following rates (from youngest to oldest): 0.1, 0.4, 0.35, 0.1, 0.05; these values come from the assumption that the disease was brought to the country from abroad, and we took that as an estimate of the age distribution of international travellers (for UK, a dataset describing age distribution of flight passengers is available and approximately equal to the provided values [28]).

- Then the number of infected individuals in each age group is split in the $E$, $I^{SC1}$ and $I^{SC2}$ compartments in the following way:

$$E_i^{in} = N_i^{in}/3, \quad I_i^{SC1,in} = \rho_i N_i^{in} \cdot 2/3, \quad I_i^{SC2,in} = (1 - \rho_i) N_i^{in} \cdot 2/3. \qquad (12)$$

The other compartments are initialized to 0, except for $S$, which is initialized to the total population in the corresponding age group obtained by the most recent country-specific census, from which the number of people seeding the infection at the start of the dynamics is subtracted.

**Parameters** which define the dynamics of our model and need to be calibrated are the following:

- $\beta$: probability of a contact between an $S$ and $I^{SC}$ individual resulting in the S individual catching the infection.

- $\kappa = 1/d_L$: transition rate of an Exposed individual becoming Infected SubClinical, with $d_L$ the average number of days in the $E$ state.

- $\gamma_C = 1/d_C$ transition rate of going from $I^{SC1}$ to $I^C$, with $d_C$ the average number of days it takes to undergo this transition.

- $\gamma_R = 1/d_R$ recovery rate from $I^{SC2}$, with $d_R$ the average number of days it takes to recover.

- $\gamma_{R,C} = 1/d_{R,C}$ recovery rate from $I^{C2}$, with $d_{R,C}$ the average number of days it takes to recover.

- $\nu = 1/d_D$ death rate from $I^{C1}$, with $d_D$ the average number of days before death occurs after entering the $I^{C1}$ state.

- $\rho_i$'s: age dependent probabilities of going to $I^C$ instead of directly recovering from the $I^{SC}$ state.

- $\rho_i'$'s: age dependent probabilities of death after being hospitalized.

- $N^{in}$: total number of individuals who carried the infection at the start of the training period (1st of March).

- $\alpha_4$: constant value of reduction in social contacts for people in age group 4, after the beginning of the lockdown period.

- $\alpha_5$: constant value of reduction in social contacts for people in age group 5, after the beginning of the lockdown period.

- $\alpha_{123}$: coefficient of proportionality between reduction of social contacts and reduction of mobility for age groups 1,2,3.

These parameters will be estimated using Approximate Bayesian Computation (ABC), which provides a posterior distribution for them—the details of the ABC methodology and associated results can be found in S1 and S2 Appendices correspondingly. The results below are integrated over this posterior distribution, which allows us to design robust controls and quantify the underlying uncertainty in our results.

## Optimised mobility values based on uncertainty

We now adopt the viewpoint of a policy maker whose task is to determine mobility restrictions on a population in order to slow down the spread of the COVID-19 epidemics, while still keeping the economic costs of lockdown as low as possible. We therefore formulate the problem in an optimal control setting. In this context, we will minimise a cost functional which includes penalties on the number of COVID-19 related deaths, hospital beds occupancy, and the economic cost of different types of lockdown.

The control variables in our problem are the reductions of the mobility values $m^{school}$, $m^{work}$, $m^{other}$ ("Mobility data"). These are related to the coefficients in the contact matrices

$\alpha^{school}$, $\alpha^{work}$, $\alpha^{other}$ via the inferred values $\alpha_{123}$, $\alpha_4$ and $\alpha_5$ in (10) and (11), relating our control policy to measurable quantities. We recall here that the mobility values $m^{loc}$, for $loc \in \{school, other, work\}$ only control the change of contacts for age groups 1,2,3 (below 60 years old), while the change of contacts for age groups 4 and 5 (above 60) is instead represented by the parameters $\alpha_4$, $\alpha_5$, which we inferred from data. Our optimisation framework therefore assumes that the reduction in social contacts for age groups 4 and 5 stays fixed to the inferred value throughout the optimisation horizon, and we optimise only on the change of mobility referred to younger age groups. This is reasonable as the older age groups constitute a minor part of the workforce and are the extremely vulnerable to the disease. Therefore, we expect that the official advice for them will be to remain with stricter isolation rules than the rest of the (working) population.

We will take advantage of the ABC inferential framework both to quantify the uncertainty of the parameters of our model and to develop a lockdown strategy that is robust to uncertainties. To this end, we proceed as follows, where the details of each step will be given throughout this section.

1. *Uncertainty quantification*: Perform inference on the model parameters using data from the public health authorities and Google mobility to obtain a posterior distribution of the parameter values given the dataset.

2. *Posterior loss based cost functional*: Define a cost functional that takes into account the economic cost of closing venues / reducing mobility to different locations and the sanitary cost of increased infection.

3. *Nonlinear model predictive control*: Optimize, over a fixed time frame, a lockdown strategy by minimizing the sanitary/economic cost functional, constrained to the inferred epidemiological dynamics. The optimisation is based on the *integrated posterior distribution*: this involves solving the epidemiological model forward using various sets of parameters values, sampled from the posterior distribution, and computing the expectation of the cost functional with respect to this distribution. This optimization step determines an optimal policy that is applied for a reduced amount of time, after which the model is updated and the optimal policy recomputed.

A diagram synthesizing this data-driven optimal control approach is presented in Fig 5. The applicability of this methodology goes beyond the design of control strategies for the COVID-19 pandemic, and can be applied to different dynamics and cost functionals.

We remark again that even though this procedure is computationally costly, it pays off by offering a control strategy that is robust to a number of possible (and highly likely) scenarios, by taking into account the uncertainty on parameter estimates [29].

## Uncertainty quantification

We use approximate Bayesian computation (ABC) [30] to calibrate the parameters of our model, by using the datasets reporting on the number of hospitalized and deceased patients released by the public health authorities:

- The daily number of deaths in hospitals attributed to COVID-19 (per age group)

- The daily number of hospitalized people with COVID-19 related diseases

We calibrate our model on data from the 1st of March up to different ending times $t_{obs}$ (eg. 31st of August). ABC is suitable for the considered task as it relies only on simulations from the model, and works by looking for values of parameters such that the integrated dynamics is

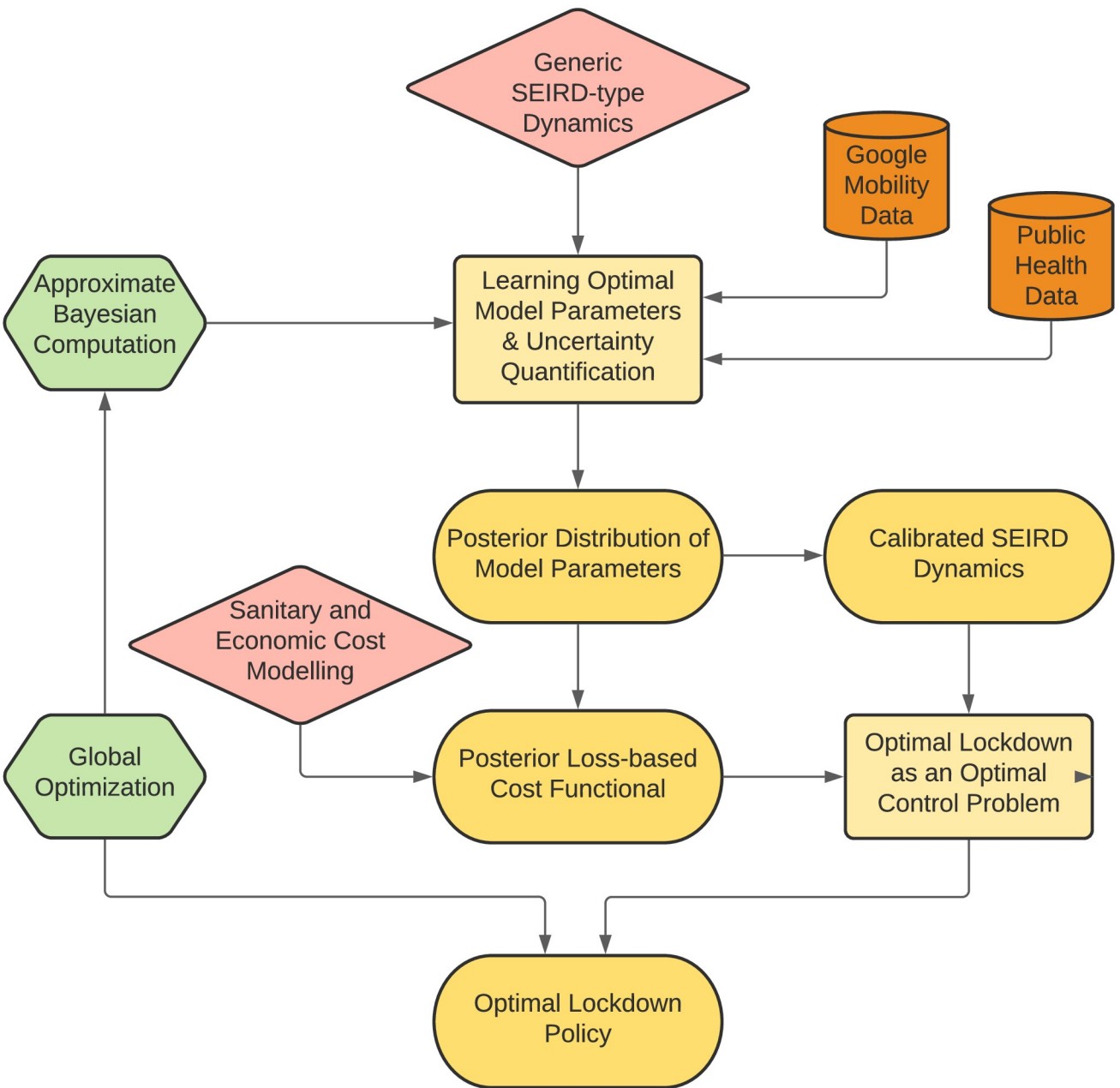

**Fig 5. Flow diagram of the data-driven optimal control approach.** Starting from a generic-type SEIRD model, we learn optimal model parameters based on mobility/healthcare datasets and Approximate Bayesian Computation. The output is a posterior distribution of model parameters, which is used to generate calibrated SEIRD dynamics and a cost functional accounting both for sanitary and economic costs of a lockdkown. These two ingredients determine the formulation of an optimal control problem, which is solved by means of a global optimization algorithm. The final output of our approach is an optimal lockdown policy which can be recalibrated as new data is fed into the system.

close to the observed one. In this way, it provides the user with samples from an approximate posterior distribution of the parameters given observed data ($\pi_{ABC}(\theta|x_{obs})$). The better the match between the simulation and the observation, the better is the approximation of the true posterior $\pi(\theta|x_{obs})$. However, this comes at higher computational cost, so that the level of approximation needs to be balanced with computational considerations. We discuss additional

details regarding the use of ABC to calibrate our model in Section 2 of S1 Appendix. Moreover, ABC allows us to fix a prior uncertainty on the values of the parameters (defined by a prior distribution $\pi(\theta)$), and to quantify prediction uncertainty through the approximate posterior $\pi_{ABC}(\theta|x_{obs})$, which we will use in designing the optimal control task next.

We remark that the ABC algorithm which we employ (discussed in Section 2 of S1 Appendix) provides us with a set of samples from the posterior distribution associated with importance weights. We can obtain independent and identically distributed (i.i.d.) samples from it by using *bootstrap*; namely, we choose (with replacement) from the set of ABC samples with probability proportional to the importance weight itself. The new set of samples generated in this way can be considered as i.i.d. from the posterior distribution of the parameters given data.

## Posterior loss based cost functional

In this step we identify the cost functional we will optimize to determine an optimal lockdown strategy. We consider a cost functional which combines the economic cost of lockdown with the sanitary cost of lifting restrictions. We focus on the representation of these terms and on the inclusion of the posterior distribution of the model parameters in its modelling. Determining the relevance of sanitary versus economic costs is a task left to the policy maker. However, when a quantitative choice has been made, our methodology allows to test the effect of such a choice and to evaluate the stability of the optimal strategy.

We denote by $t_0$ the start of the optimisation interval, corresponding to the day we wish to start the new lockdown policy (in our first example, the 24th of May), and by $T_h$ the length of the interval in days, for which we want to apply the lockdown strategy, also known as optimisation horizon. To study the cost of lifting restrictions, we penalise the predicted number of deaths during the optimisation horizon $[t_0, t_0 + T_h]$. This is given by

$$\sum_i D_i(t_0 + T_h) - D_i(t_0) = \sum_{t=t_0+1}^{t_0+T_h} \sum_i \Delta D_i(t) \,,$$

where $\Delta D_i(t)$ is the daily increase in the number of deceased in the age group $i$. Furthermore, we need to guarantee that the number of infected individuals who need hospitalisation, $I^C = \sum_i I_i^{C_1} + I_i^{C_2}$, remains below the overall hospitals capacity $H_{max}$. This could be included as a hard state constraint, here instead we penalise the event in which $I^C$ surpasses the total capacity by including a term of the form

$$\Phi(I^C) := \max(I^C - H_{max}, 0) \,.$$

We point out that this term will not be activated if the levels of infected people who need hospitalisation remain well below $H_{max}$, which here we take to be $H_{max} = 10000$. For example, the number of people in hospital in England with COVID-19 related symptoms at the end of our first training window (23rd of May) was 7106, while at the height of the peak of the first wave of the epidemic in England (on the 12th of April) this number was 17933. For France, numbers reach higher values; however, we keep the same value of $H_{max} = 10000$, in order to penalize a large number of infected people in the same way across the two countries, and to have comparable values in the optimization strategy. As a final measure of the sanitary cost, we introduce a final time cost, where we penalise the basic reproduction number ($\mathcal{R}$) at the end of the optimisation horizon $\mathcal{R}(t_0 + T_h)$ (details about $\mathcal{R}$ number for our model can be found in Section 1 of S1 Appendix). This terminal penalty ensures that the control strategy does not simply output an optimal solution which switches off the reduction on mobility towards the end of the

optimisation horizon. While such a solution is consistent with the optimal control design and is an interesting instance of the turnpike phenomenon [31], it is not suitable in our context. Finally, to account for the economic cost of lockdown, we penalise the mobility reduction by introducing a quadratic cost of the form $\|\mathbf{1} - m^{loc}(t)\|^2$, for $loc \in \{school, other, work\}$.

As mentioned before, the forward model is run for different sets of parameters drawn from the posterior distribution obtained via ABC inference. For this reason, the $\Delta D_i(t)$ and $I_i^C(t)$ variables appearing in the cost functional depend on the chosen value of the parameters of the model. We hide the explicit dependence for notational convenience. The cost is computed by taking an expectation over the posterior distribution for the parameters, since each parameter value will lead to a different realisation of the dynamics. Due to the nonlinearity of the system, this is clearly not the same as minimizing the objective using the posterior mean of the parameters only. In practice, we use 50 i.i.d. samples from the posterior distribution and approximate the expectation with an average over the trajectories obtained with those parameter values. Minimizing an expected cost in this way is much more computationally expensive than a cost computed on a point estimate of the parameters (for instance the posterior mean). However, it provides a way to take into account the uncertainty in the parameters while producing more robust results.

Collecting the different terms in our cost, we optimise

$$
\min_{m(\cdot)\in\mathcal{M}} \mathcal{J}(m) := \sum_{t=t_0+1}^{t_0+T_h} \left[ \frac{1}{2}\mathbb{E}\left[ \sum_i \Delta D_i(t) + \Phi(I^C(t)) \right] \right.
$$
$$
\left. + \sum_{loc\in\{school,\ other,\ work\}} \frac{\epsilon_{loc}}{2}\|\mathbf{1} - m^{loc}(t)\|^2 \right] + \mathbb{E}[\mathcal{R}(t_0 + T_h)],
$$

(13)

through the control signal

$$
m(t) = (m^{school}(t), m^{work}(t), m^{other}(t)) \in \mathcal{M} := \{m : [t_0, t_0 + T_h] \to [0,1]^3\},
$$

where $\epsilon_{loc}$, for $loc \in \{school, other, work\}$, represents the relative cost of limiting the mobility to schools, workplaces and other locations with respect to the sanitary cost, and where the expectation is taken over the posterior distribution of the parameters of the model. The choice of the values for $\epsilon_{loc}$ affects the optimal policy by attributing a larger economic or social cost of closing one of the categories with respect to the others. Determining adequate weights for these costs is the ultimate task of the policy maker.

## Nonlinear model predictive control

To close our optimal control formulation we add specifications to the controls we expect to obtain, restricting the space of admissible signals. As the values of $\alpha$ are between 0 and 1, a reasonable assumption is to expect the $m^{loc}$ to be in the interval [0, 1] as well, for $loc \in \{school, work, others\}$. However, due to the fact that we cannot impose a 100% closure of all settings, we set a lower bound for $m^{loc}$ to be the *lowest value* of each $m^{loc}$ observed during the lockdown period. This results in the constraints $m^{work} \in [0.31, 1]$, $m^{school} \in [0.1, 1]$, and $m^{others} \in [0.41, 1]$.

As can be seen from Eq (13), the controls $m^{loc}$, $loc \in \{school, work, others\}$ are time-dependent, and they are computed by minimising the cost functional (13) subject to the state constraints (1)–(8). Ideally, the numerical realization of the optimal control strategy would be driven by the calculation of first-order optimality conditions and a reduced gradient approach to minimise $\mathcal{J}(m)$. However, the nonlinearities in the dynamics and in the terminal penalty,

where the reproduction rate is expressed as an eigenvalue of a parameter-dependent matrix, make our problem highly non-convex. Moreover, the penalty $\Phi(I^C)$ is non-differentiable. For the purposes of this paper we will compute the optimal control by using generalized simulated annealing (or dual annealing) [32]. The use of meta-heuristics for the solution of large-scale nonlinear optimal control problems has been assessed in [33, 34]. We embed the solution of the optimal control problem (13) in a nonlinear model predictive control (NMPC) framework [35]. To this end, we select a prediction horizon $T_{opt}$, and optimise the control variables $m^{loc}(t)$, $t \in [t_0, t_0 + T_{opt}]$ using the current state at $t_0$ as our initial condition. From the optimal control sequence we recover the optimal action for a single day, that is $m^{loc}(t)$, $t \in [t_0, t_0 + 1]$ and evolve the dynamics for the same amount of time, and repeat the optimisation procedure in the updated time frame $[t_0 + 1, t_0 + 1 + T_{opt}]$. This process is then repeated until the complete optimisation interval $[t_0, t_0 + T_h]$ is covered. The NMPC methodology recovers a robust optimal control law in feedback form that can be adjusted to account for disturbances in the control loop. Therefore, instead of using the current state predicted by the model as initial condition, we can update this to be the current state of the population in the considered country estimated from data, every time new data becomes available. This ensures that the control methodology accounts for noisy observations, or for unexpected variations in the data.

## Results and discussion

In this section, we apply our methodology to each of the datasets specified in the previous sections. In the first half of the section, as a proof of concept, we show the importance of the various terms included in the cost functional and their influence on the results, and this is done for the England dataset with parameter values calibrated with data between the 1st of March and the 23rd of May, with lockdown strategies applied for 90 or 120 days starting on the 24th of May. In the second half, we apply our methodology to two populations: England and France, with the models calibrated up to the 31st of August and lockdown strategies applied from the 1st of September. The posterior distributions of the parameters are available in S2 Appendix. This section is organised as follows:

1. Choice of an appropriate prediction horizon, $T_{opt}$.

2. Influence of the relative weight of the economic and sanitary costs (i.e., how large to choose each of the $\epsilon_{school}$, $\epsilon_{work}$ and $\epsilon_{other}$).

3. Influence of the relative costs between reducing mobility to different locations (i.e., the relative weights of $\epsilon_{school}$, $\epsilon_{work}$ and $\epsilon_{other}$).

4. Dynamic update of the control strategy as we recalibrate the model.

5. Application of our methodology to England and France.

Steps 1–3 can inform a policy maker on their decision of how to weight each term in the cost functional, but we remark that this is ultimately their decision. In steps 4 and 5 we fix the parameters explored in 1–3 and test the methodology in specific cases.

Before presenting the optimal lockdown results, we exemplify our inference results for England with plots of the variables that are relevant for the control methodology—for the complete inference results, we refer the reader to S2 Appendix. In Fig 6, we plot the predicted number of people in the $I^C$ category—the red line is the median prediction and the shaded area denotes a confidence interval with 99% credibility—compared to the true data in green in the left panel. The middle panel reports on the same results for the number of deceased people, and the right panel reports the predicted basic reproduction number, $\mathcal{R}(t)$.

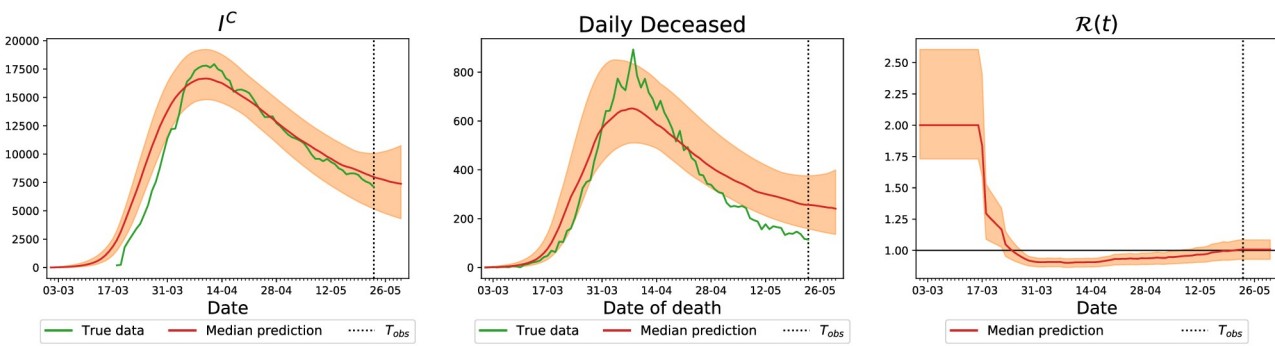

**Fig 6. Comparison of predictions of our model with the real number of hospitalized people with COVID-19 and total daily deaths (green) on 23rd May for England.** The solid red line denotes the median prediction, filled spaces denote the 99% credible interval and the vertical dashed line denotes the observation horizon. The different columns represent number of people in hospital, $I^C$ (left), daily deceased (middle) and value of $\mathcal{R}(t)$ (right).

## Proof of concept of the methodology

We now proceed to present the results of our methodology as described above on data for England observed until 23rd May. Each of the figures below is either of two types: 1) Lockdown strategy and its effect on the number of hospitalized individuals, and 2) Influence of the lockdown strategies on the reproduction number $\mathcal{R}(t)$. In the first case, we plot the number of hospitalised people (full red line) for each control strategy—these results are associated with the red axis, on the left of the figure—and the value of the optimized mobility values $m^{loc}$ for $loc \in \{school, work, others\}$ in the three blue lines, associated with the blue values on the right axis. The full, dashed, and dash-dotted lines correspond to work, school and other settings, respectively.

**Dependence on $T_{opt}$.** A crucial issue in the NMPC framework is the selection of the prediction horizon $T_{opt}$. For small prediction horizons, the optimal action tends to be instantaneous and loses its capability to foresee long-term consequences of the policy. On the other hand, a sufficiently long prediction horizon will enforce a stabilizing control law, but its numerical realization becomes increasingly complex. Therefore, at the core of the selection of a suitable prediction horizon there is a trade-off between short-sightedness, stabilization capabilities of the policy, and computability. This is exemplified in Fig 7, where we illustrate the role that is played by the prediction horizon in the performance of the control loop for a fixed value of $\epsilon_{loc}$, for $loc \in \{school, work, others\}$. It can be observed that a short-sighted policy, with

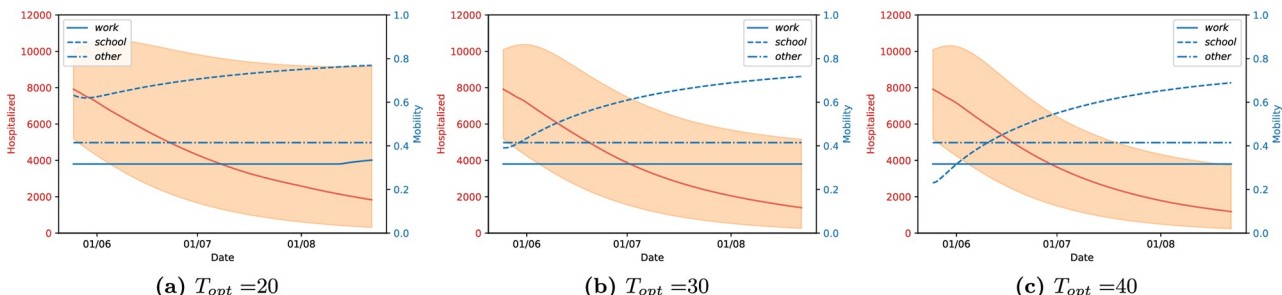

**(a)** $T_{opt} = 20$ **(b)** $T_{opt} = 30$ **(c)** $T_{opt} = 40$

**Fig 7. Dependence on the prediction horizon $T_{opt}$ for determining and optimal control strategy for England.** Here, $(\epsilon_s, \epsilon_w, \epsilon_o) = (100, 100, 100)$ and the lockdown strategies were applied for $T_h$ = 90 days starting on the 24th of May.

a prediction horizon of 20 days (left panel), is less stringent in the mobility reduction, causing a larger number of hospitalized people in the long term and a large uncertainty in the end result.

However, it is clear that the lockdown strategies are similar for both prediction horizons of $T_{opt}$ = 30 (middle panel) and $T_{opt}$ = 40 (right panel): here, we observe a smaller overall number of hospitalized people over time, while the uncertainty on the results remains bounded, with slightly smaller credibility intervals in the second case. For this reason, we will from now on use the prediction horizon of $T_{opt}$ = 30, which is long enough to avoid a short-sighted strategy, while being short enough to be computationally cheaper. We point out that this is correctly aligned with the COVID-19 time scale for transmission, is a reasonable time frame for a policy maker, and agrees with PHE definition of death due to COVID, where a death is considered COVID-related if it is within 28 days of a positive COVID test [36].

**Influence of the relative weights of the control penalties ($\epsilon_{school}$, $\epsilon_{work}$, $\epsilon_{other}$).** The choice of suitable control penalties $\epsilon_{loc}$, $loc \in \{school, work, other\}$ is a sensitive issue in any optimal control problem, and we now proceed to study its effect on the resulting lockdown strategies. In this context, there are two important properties to explore: the relative weight between the sanitary and economic costs of our lockdown strategy, which is represented by how large the values of each of $\epsilon_{school}$, $\epsilon_{work}$, $\epsilon_{other}$ are, and the relative importance between each of the $\epsilon_{school}$, $\epsilon_{work}$, $\epsilon_{other}$. We observe that the predicted number of daily deceased at the end of the training interval is between 200 and 400 (see Fig 6). Having in mind that the control variables are constrained to [0, 1], we conclude that each $\epsilon_{loc}$ should be in the order of 100, so that the control strategies are sensitive to both the sanitary and economic costs.

In our next example, we assume that the economic cost of opening each type of location is the same (i.e. $\epsilon_{school} = \epsilon_{work} = \epsilon_{other}$) and analyse the effect of varying their relative weight to that of the sanitary cost. This is shown in Fig 8, where we present the lockdown strategies and corresponding values of hospitalized people for $\epsilon_{loc}$ = 100 (left) and $\epsilon_{loc}$ = 200 (right), for $loc \in \{school, work, other\}$. We observe that higher values of $\epsilon_{loc}$ result in the strategies which open workplaces earlier, as keeping them closed is more expensive, but also result in higher uncertainty. Interestingly, however, we observe that both control strategies keep $m^{other}$ constant at a value of approximately 0.41, which is the minimum value allowed for this parameter. This behaviour is consistently reproduced for any of the values $\epsilon_{loc}$ that we explored as we will see below; the only situation in which we did not observe this behaviour was by attributing an unreasonably high weight to the economic cost, in which case all the lockdown measures are lifted, leading to a large increase of the number of infected people and to a second wave of the

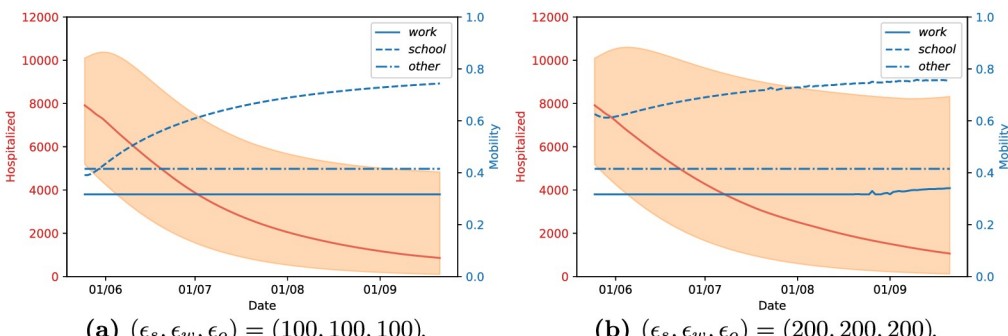

**(a)** $(\epsilon_s, \epsilon_w, \epsilon_o) = (100, 100, 100)$.　　　　**(b)** $(\epsilon_s, \epsilon_w, \epsilon_o) = (200, 200, 200)$.

**Fig 8. Different relative weights between the sanitary cost and economic cost of the lockdown measures with the lockdown strategies starting on the 24th of May and applied for $T_h$ = 120 days, for England.**

epidemics. To better understand this phenomenon, we inspect the contact matrix relative to the "other locations" category (Fig 3). There, it can be observed that the setting in which there are more contacts between the part of the population in which our optimisation strategy acts, (under 60 year-olds), and the older population is the "other locations". As the latter are the most vulnerable to the disease, the optimisation is limiting the number of contacts they have by reducing $m^{other}$.

The last property we investigate is the effect of varying the values of $\epsilon$ for each category, which can produce a more economically viable solution. In the cases we will explore, we observe that opening workplaces is clearly an important part of the economy, and this might not be achievable without opening schools as well. For this reason, we will attribute the highest economic costs to closing workplaces, followed by schools and then others. We point out that this is ultimately a choice of the policy maker, and in principle any combination of values for each $\epsilon_{loc}$ could be considered. We tested a variety of combinations of these cost weights, each providing us with a lockdown strategy. In this case, we compare the efficiency of each strategy in Figs 9 and 10. As can be seen in Fig 9, all of the strategies propose to open workplaces and schools earlier or later (or not at all, in the case of schools) depending on the relative weights between sanitary and economic costs, and, as before, keep other locations at its minimum value. We show an additional comparison in Fig 10, where we plot the reproduction number $\mathcal{R}(t)$ resulting from each strategy, compared with what it would be if the mobility values remained unchanged from their estimated values on the 29th of May. We note that the optimal strategies always keep the value of $\mathcal{R}$ smaller than 1 for most of the optimization horizon, but the confidence intervals allow for values larger than 1 at the end of the optimization window.

## Dynamic update of the model and optimal control strategy

As new data becomes available, we can re-perform the model fit, and obtain a new posterior distribution on the parameters to use in order to find the optimal mobility values. An instance

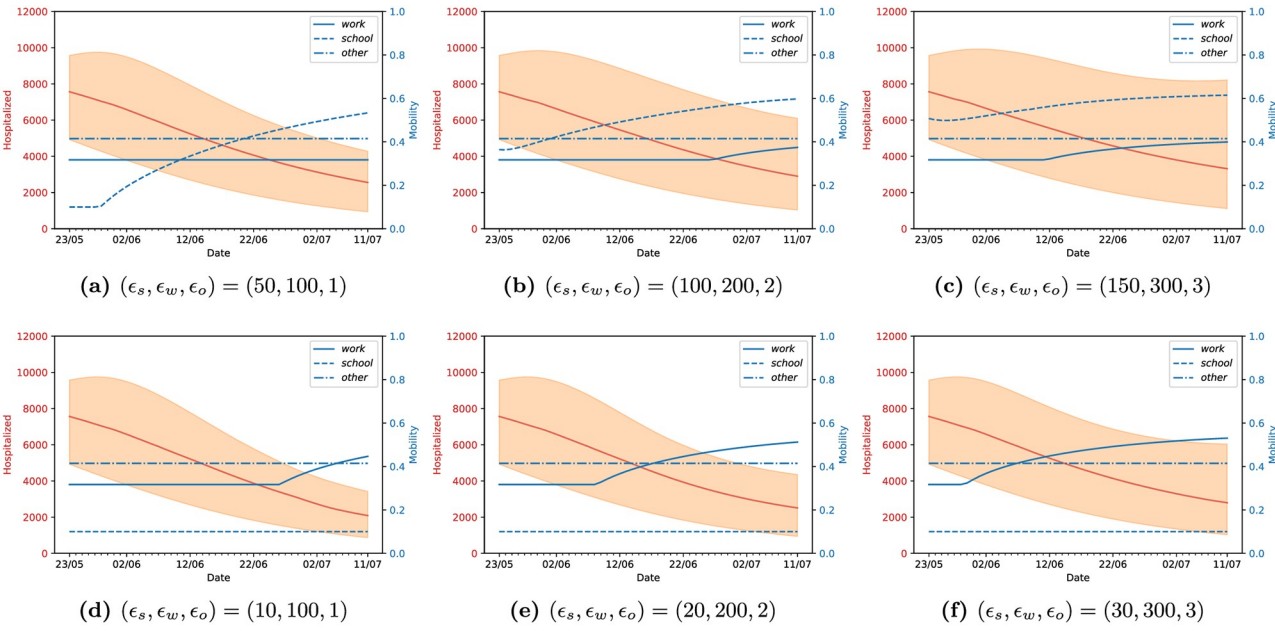

**(a)** $(\epsilon_s, \epsilon_w, \epsilon_o) = (50, 100, 1)$ **(b)** $(\epsilon_s, \epsilon_w, \epsilon_o) = (100, 200, 2)$ **(c)** $(\epsilon_s, \epsilon_w, \epsilon_o) = (150, 300, 3)$

**(d)** $(\epsilon_s, \epsilon_w, \epsilon_o) = (10, 100, 1)$ **(e)** $(\epsilon_s, \epsilon_w, \epsilon_o) = (20, 200, 2)$ **(f)** $(\epsilon_s, \epsilon_w, \epsilon_o) = (30, 300, 3)$

**Fig 9. Different economic cost weights produce different opening strategies and predicted hospitalized people, with lockdown strategies starting on the 24th of May and applied for $T_h$ = 120 days, for England.**

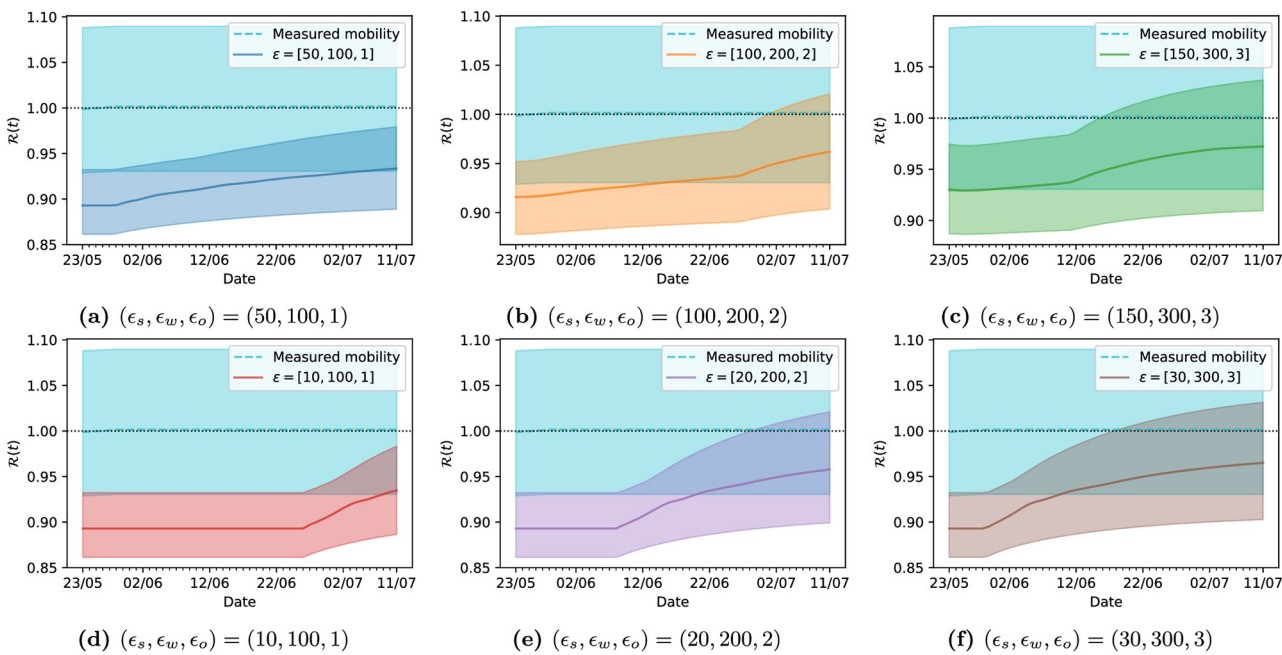

**Fig 10. Evolution of $\mathcal{R}(t)$ corresponding to different opening strategies starting on the 24th of May, learned using different economic cost weights for different $\epsilon$ values, with 99% credibility intervals; these plots are referred to England.**

of this *dynamic update* can be seen in Fig 11, where we show the optimal mobility values and the corresponding basic reproduction number after the model parameters and the optimal solution are updated at four instances: the 11th of April, 26th of April, 11th of May and 23rd of May. Here we used $(\epsilon_{school}, \epsilon_{work}, \epsilon_{other}) = (150, 300, 3)$ as from the results in Figs 9 and 10 it can be seen that such a choice leads to a lockdown strategy that increases the mobility towards both workplaces and schools while keeping the value of $\mathcal{R}(t) < 1$.

We observe that the lockdown strategy determined with parameter values fitted on data up to the 11th of April is extremely restrictive, as the predicted dynamics on that date badly over-estimates the number of deceased and hospitalized people (see S2 Appendix). However, we see

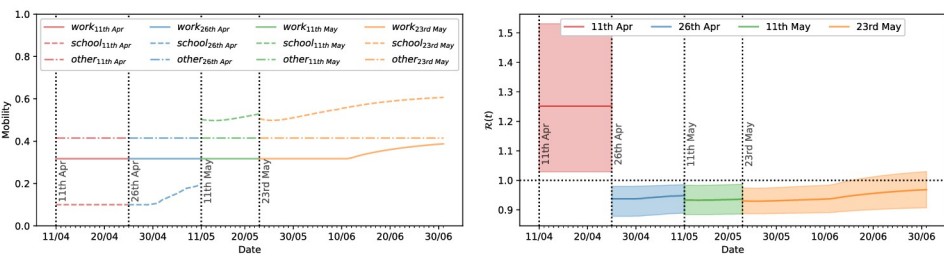

**(a)** Dynamically updated control strategy.

**(b)** $\mathcal{R}(t)$ corresponding to the updated control strategies.

**Fig 11. Dynamically updated control strategy: We fit the model on data for England up to $t_{obs}$ = 11th of April and determine the optimal mobility strategy up to the next observation point $t_{obs}$ = 26th April.** Data until the latter is used to repeat the procedure, in order to update the optimal control strategy exploiting newly available information. Here $(\epsilon_{school}, \epsilon_{work}, \epsilon_{other}) = (150, 300, 3)$. In panel (a), we show the resulting optimal mobility values, with the corresponding values of $\mathcal{R}(t)$ and their credibility intervals shown in panel (b).

that recalibrating the parameters and updating the lockdown strategy using newly available data is beneficial: using data until the 26th of April results in a better prediction of hospitalized and deceased numbers, and we see an increase of the mobility towards schools already at the end of April. We repeat the procedure on the 11th and 23rd of May, resulting on a proposed strategy allowing schools to be open even further at the first instance, and then remaining almost unchanged, with an increased mobility towards workplaces around mid-June. We also note that, with the exception of the time interval between the 11th and the 26th of April— where the high value of $\mathcal{R}(t)$ is due to the fact that for the current values of the parameters the epidemic is predicted to increase rapidly, independently of the lockdown measures—the predicted $\mathcal{R}(t)$ value always stays below 1. This dynamic update of the model allows us to rely less on long-term predictions from the model, which are more computationally expensive and become more biased the farther in the future the prediction is. Together with the knowledge on model-specific biases coming from fitting the model to different horizons in the past, this approach enables a policy maker to assess the validity of the proposed optimal mobility strategy and update it on a periodic basis as new data becomes available.

## Optimal strategy for England and France on 1st of September

After exploring all the different properties of our optimal control strategy, we proceed to test it in two populations, England and France, in a more recent setting, which is closer to a second wave of the epidemics. We fit the model again with data from the PHE and the NHS for England, and from SpF for France, and using the Google mobility data in both cases. The inference was performed on both cases for data up to the 31st of August and the results are presented in Fig 12 below.

**Optimal lockdown strategy for England starting in September 2020.**  We apply our optimal control strategy again for the population of England, for a more recent time interval. As mentioned above, the parameters were fitted up to the 31st of August, and we apply a lockdown strategy for $T_h$ = 120 days, starting on the 1st of September. The prediction horizon was kept at $T_{opt}$ = 30 days and we used two possible sets of values for $\epsilon_{loc}$. The corresponding results are shown in Fig 13, where the first two panels show the number of hospitalized people and corresponding credibility intervals (full red line and dashed area) and the mobility values (blue lines), while the corresponding values of $\mathcal{R}(t)$ (and credibility intervals) are presented in the right panel.

**Optimal lockdown strategy for France starting in September 2020.**  Our methodology is valid in general, and in particular it can be applied to different datasets. Our last example applies the strategy to the French population, where the epidemics has exhibited a different evolution than in the UK. As in the previous paragraph, we fitted the parameters with data up to the 31st of August, and apply the lockdown strategy for $T_h$ = 120 days, starting on the 1st of September. The prediction horizon was kept at $T_{opt}$ = 30 days and we used two possible sets of values for $\epsilon_{loc}$. The corresponding results are shown in Fig 14, and are organised in a similar manner to Fig 13. We observe that in this case the relative values of $\epsilon_{loc}$ have a much stronger influence, with the lockdown strategy shown in the left panel having extremely better results than that on the middle panel. This is visible both on the number of hospitalized people and the resulting reproduction number, and illustrates the importance of the choice of these parameters and associated discussion with policy makers.

We can see a significant difference in the data reported in these two countries (green lines in Fig 12) which leads to very different parameter estimates of the epidemic model (see S2 Appendix). This is reflected in the optimal control we inferred for these two countries, highlighting the adaptability of our methodology for different countries and different

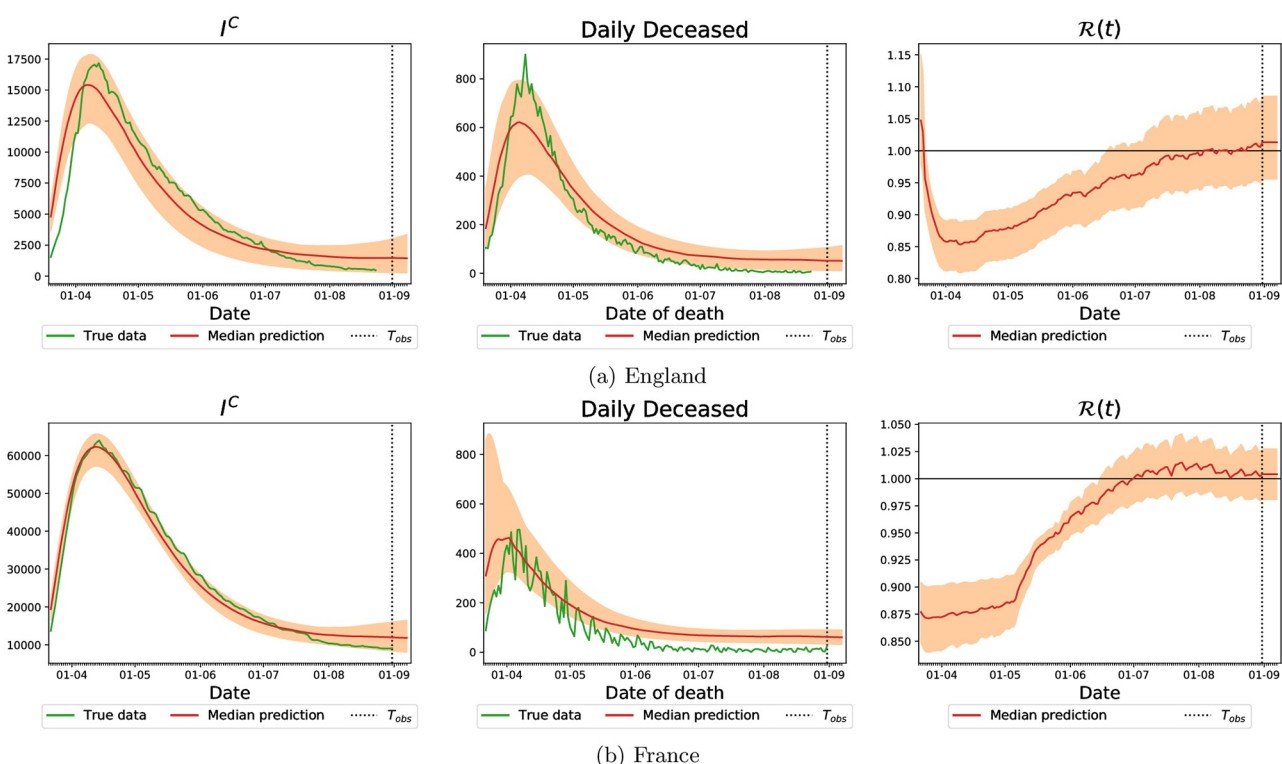

(a) England

(b) France

**Fig 12. Comparison of predictions of our model with the real number of hospitalized people with COVID-19 and total daily deaths (green) on 31st August in England and France.** The solid red line denotes the median prediction, filled spaces denote the 99% credible interval and the vertical dashed line denotes the observation horizon. The different rows represent different observation horizons, while the columns represent number of people in hospital ($I^C$ compartment, left column), daily deceased (middle column) and value of $\mathcal{R}(t)$ (right column).

scenarios. We reiterate that our methodology for both England and France recommends to keep mobility of "others" as low as possible if not at the lowest possible label of full lockdown. Remembering that the others category refers to "retail", "transit" or "grocery" locations, our methodology is recommending that the we should keep a stringent lockdown until December, while opening up the work and school places in a socially-distanced manner. Deviations from that will likely result in a second-wave of infection spreading.

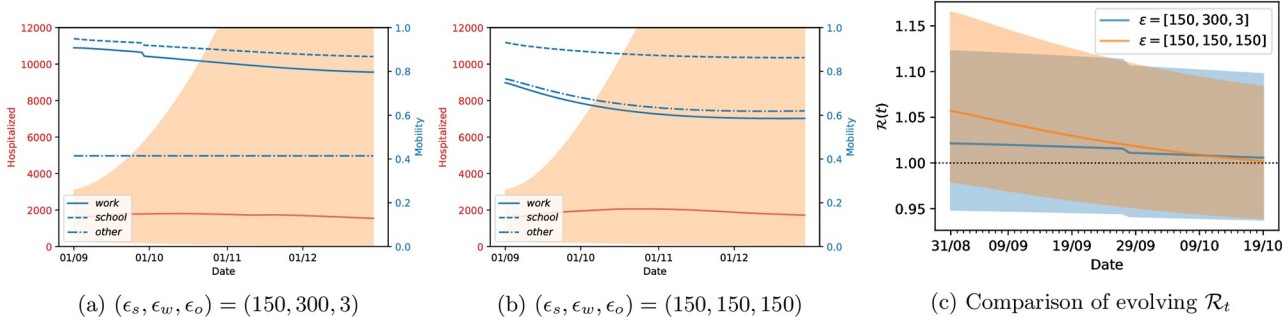

(a) $(\epsilon_s, \epsilon_w, \epsilon_o) = (150, 300, 3)$

(b) $(\epsilon_s, \epsilon_w, \epsilon_o) = (150, 150, 150)$

(c) Comparison of evolving $\mathcal{R}_t$

**Fig 13. Optimal strategy for England with parameters fitted with data up to the 31st August.** The lockdown strategy is applied for 120 days starting on the 1st of September and uses a prediction horizon of $T_{opt} = 30$ days.

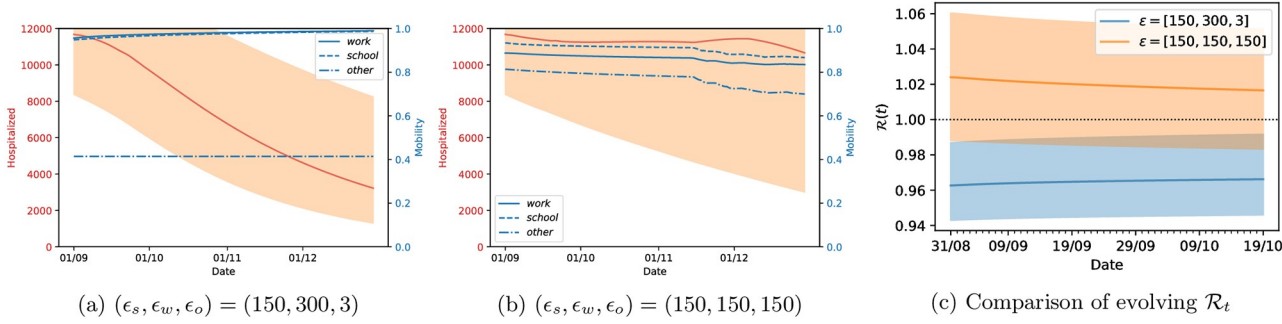

(a) $(\epsilon_s, \epsilon_w, \epsilon_o) = (150, 300, 3)$    (b) $(\epsilon_s, \epsilon_w, \epsilon_o) = (150, 150, 150)$    (c) Comparison of evolving $\mathcal{R}_t$

**Fig 14. Optimal strategy for France with parameters fitted with data up to the 31st August.** The lockdown strategy is applied for 120 days starting on the 1st of September and uses a prediction horizon of $T_{opt} = 30$ days.

## Concluding remarks and future work

We have proposed a general estimation/control methodology for the determination of optimal lockdown strategies in the context of the COVID-19 pandemic. Our approach is composed of the following elements: system dynamics described through an age-structured SEIRD model, the use of public data such as Google Mobility for estimating model parameters, and the design of adaptive lockdown strategies in the framework of nonlinear optimal control. The current work focused on a study of the COVID-19 pandemic in England and France, however the underlying methodology can be extended to other spatio-temporal locations. It can be applied to epidemiological models in general, assuming the availability of healthcare and mobility data for a suitable calibration of the dynamics. Our systematic approach provides a computational tool to assist the design of lockdown strategies which can be periodically rectified as the model is fed with incoming public data. Moreover, the proposed strategies are parsimonious in the sense that they encode both healthcare and socio-economic factors, and realistic as they are expressed as mobility reduction parameters which can be effectively measured, as opposed to switching on-off strategies.

The control strategy our framework devises strictly depends on the quality of the forecast for the evolution of the epidemics. Therefore, we expect improvements in the former can be obtained by upgrading the epidemiological model. For instance, in some of the considered scenarios our model overestimates the number of deaths and hospitalized people at the end of the prediction horizon (as for instance in Fig 6), and this could lead to a conservative suggested control strategy (although we believe that, when it comes to human lives a more conservative approach is to be preferred). This discrepancy may be caused by our model not taking into account the increased capacity of the health system to fight the disease through better treatments and more extensive testing. Specifically, it is reasonable to assume that a larger proportion of people in the subclinical compartment get tested as the epidemics progresses; ideally, these people will adhere to stringent social isolation regimes with the expectation to slow down the spread of the disease. However, how the latter affects the evolution of the epidemics when reduced mobility measures for all citizens are already in place remains unclear.

As we continue to work on our approach, a natural way to improve the accuracy of our model is through a further downscaling of our dynamics, for instance by considering 418 principal local authorities (LA) in the UK, along with a commuting network of UK citizens between them constructed from the 2011 census data. Such a refined model, whose numerical treatment will necessarily require the use of high-performance computing resources, would allow the design of space-time adaptive lockdowns. Our progress along these lines will be

documented on the companion website [37]. Code implementing the described experiments is available at https://github.com/OptimalLockdown.

## Supporting information

**S1 Appendix. Additional mathematical details.** Details on computation of the basic reproduction number $\mathcal{R}(t)$ and explanation of Approximate Bayesian Computation inferential procedure.
(PDF)

**S2 Appendix. Additional experimental results.** Inference results on the posterior distribution of parameters and on the evolution of the unobserved compartments, using data until the 23rd of May; additional optimal control results using data until 11th May.
(PDF)

## Author Contributions

**Conceptualization:** Ritabrata Dutta, Susana N. Gomes, Dante Kalise, Lorenzo Pacchiardi.

**Data curation:** Ritabrata Dutta, Lorenzo Pacchiardi.

**Formal analysis:** Ritabrata Dutta, Susana N. Gomes, Dante Kalise, Lorenzo Pacchiardi.

**Investigation:** Ritabrata Dutta, Susana N. Gomes, Dante Kalise, Lorenzo Pacchiardi.

**Methodology:** Ritabrata Dutta, Susana N. Gomes, Dante Kalise, Lorenzo Pacchiardi.

**Software:** Ritabrata Dutta, Lorenzo Pacchiardi.

**Validation:** Ritabrata Dutta, Lorenzo Pacchiardi.

**Visualization:** Ritabrata Dutta, Lorenzo Pacchiardi.

**Writing – original draft:** Ritabrata Dutta, Susana N. Gomes, Dante Kalise, Lorenzo Pacchiardi.

**Writing – review & editing:** Ritabrata Dutta, Susana N. Gomes, Dante Kalise, Lorenzo Pacchiardi.

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
