## [Editor Report · Decision Letter 0]

10 Aug 2020

Dear Dr Dutta,

Thank you very much for submitting your manuscript "Using mobility data in the design of optimal lockdown strategies for the COVID-19 pandemic in England" for consideration at PLOS Computational Biology.

As with all papers reviewed by the journal, your manuscript was reviewed by members of the editorial board. In light of the reviews (from the editor, see below), we would like to invite the resubmission of a significantly-revised version that takes into account the editors' comments.

We cannot make any decision about publication until we have seen the revised manuscript and your response to the editor's comments. Your revised manuscript is also likely to be sent to reviewers for further evaluation.

Sincerely,

Virginia E. Pitzer, Sc.D.

Deputy Editor

PLOS Computational Biology

Virginia Pitzer

Deputy Editor

PLOS Computational Biology

I found this to be an interesting and comprehensive analysis. My recommendation, at this point, is not to send this out for further review. In it's current form, I am concerned that this work will not be well reviewed and I would encourage the authors to take some additional time to develop the manuscript further and consider a re-submission after considerable revision. The analysis presented herein is both comprehensive and complex. But, at present, it reads more as a technical report with limited scope to the explicit setting that is presented. I would encourage the authors to take some time to restructure the presentation of the results and discussion, in particular, to highlight the generality of this work and the contribution in the light of the other work in the field. The authors do take some time to compare the work to previous efforts in the introduction, but this should be revisited in the discussion after the reader has digested the work that is presented. For many readers, this work will be overwhelming given the many technical areas that are overlapping: disease surveillance, age structured models, CDRs, ABC, and optimal control. The intersection of these many approaches is a real strength of this work, in my opinion, but that also requires that additional effort be taken to communicate how these pieces come together in the synthesis of the work. The optimal control component of this is really interesting and may also be the most heavily scrutinized because of the need to make explicit value judgements in the objective function (the authors acknowledge in the methods that this will be the responsibility of the decision-maker). I would encourage the authors to take more time in the discussion to go through the implications of this work and highlight whether they want the reader to take away from this a message about the optimal policies for the UK or a template for approaching the complex problem of designing optimal control strategies. If the latter, the results should be structured to highlight the benefits and challenges of this approach and the discussion should attempt to translate those in less technical detail.

I would encourage the authors to further develop this work and I would look forward to the opportunity to review a revised manuscript.
---

## [Decision Letter · Decision Letter 1]

10 Feb 2021

Dear Dr Dutta,

Thank you very much for submitting your manuscript "Using mobility data in the design of optimal lockdown strategies for the COVID-19 pandemic" for consideration at PLOS Computational Biology. As with all papers reviewed by the journal, your manuscript was reviewed by members of the editorial board and by several independent reviewers. The reviewers appreciated the attention to an important topic. Based on the reviews, we are likely to accept this manuscript for publication, providing that you modify the manuscript according to the review recommendations.

We have received two complementary reviews for the revised manuscript. Both reviewers raise a few minor points that should be considered in a revision -- specifically some clarification on notation, inclusion of references to ABC methods, and clarification about assumptions. Please address these small changes and submit a revised manuscript. I will be able to make a final determination without additional review by the referees.

Note that R2's comments appear to reference an alternate page/line numbering system: below I have identified a few specific points to address with the number system of the original document.

1. Page5/20 - clarify the sentence "we assume the ones for England are well represented by those". Perhaps just simplify to "We use the contact matrix for England is the same as that for the whole of the UK "

2. Figure 4 legend -- add "we" before "note"

3. L489 - provide some justification for the claim that the 30-day prediction horizon is "aligned with the COVID-19 time scale for transmission" (serial interval is much smaller than 30 days.)

Sincerely,

Matthew (Matt) Ferrari

Associate Editor

PLOS Computational Biology

Virginia Pitzer

Deputy Editor-in-Chief

PLOS Computational Biology

[LINK]

We have received two complementary reviews for the revised manuscript. Both reviewers raise a few minor points that should be considered in a revision -- specifically some clarification on notation, inclusion of references to ABC methods, and clarification about assumptions. Please address these small changes and submit a revised manuscript. I will be able to make a final determination without additional review by the referees.

Note that R2's comments appear to reference an alternate page/line numbering system: below I have identified a few specific points to address with the number system of the original document.

1. Page5/20 - clarify the sentence "we assume the ones for England are well represented by those". Perhaps just simplify to "We use the contact matrix for England is the same as that for the whole of the UK "

2. Figure 4 legend -- add "we" before "note"

3. L489 - provide some justification for the claim that the 30-day prediction horizon is "aligned with the COVID-19 time scale for transmission" (serial interval is much smaller than 30 days.)

Reviewer's Responses to Questions

**Comments to the Authors:**

Reviewer #1: find my comment in the attachment

Reviewer #2: See the attached file

**Have all data underlying the figures and results presented in the manuscript been provided?**

Reviewer #1: None

Reviewer #2: None

PLOS authors have the option to publish the peer review history of their article (what does this mean?). If published, this will include your full peer review and any attached files.

Reviewer #1: No

Reviewer #2: No
---

## [Decision Letter · Decision Letter 2]

2 Jul 2021

Dear Dr Dutta,

We are pleased to inform you that your manuscript 'Using mobility data in the design of optimal lockdown strategies for the COVID-19 pandemic' has been provisionally accepted for publication in PLOS Computational Biology.

Best regards,

Matthew (Matt) Ferrari

Associate Editor

PLOS Computational Biology

Virginia Pitzer

Deputy Editor-in-Chief

PLOS Computational Biology

Reviewer's Responses to Questions

**Comments to the Authors:**

Reviewer #1: I believe the authors have answered to all my concerns and the paper can be published in PLOS computational biology as is.

Reviewer #2: Check the reference style since there are many references cited in different formats

Check and uniform the style of the list of parameters you made at page 8

**Have the authors made all data and (if applicable) computational code underlying the findings in their manuscript fully available?**

Reviewer #1: Yes

Reviewer #2: None

PLOS authors have the option to publish the peer review history of their article (what does this mean?). If published, this will include your full peer review and any attached files.

Reviewer #1: No

Reviewer #2: No

---

## [Editor Report · Acceptance letter]

19 Jul 2021

PCOMPBIOL-D-20-01162R2 

Using mobility data in the design of optimal lockdown strategies for the COVID-19 pandemic

Dear Dr Dutta,

I am pleased to inform you that your manuscript has been formally accepted for publication in PLOS Computational Biology. Your manuscript is now with our production department and you will be notified of the publication date in due course.

With kind regards,

Katalin Szabo
